# Non-Anticoagulant Activities of Low Molecular Weight Heparins—A Review

**DOI:** 10.3390/ph16091254

**Published:** 2023-09-05

**Authors:** Ke Feng, Kaixuan Wang, Yu Zhou, Haoyu Xue, Fang Wang, Hongzhen Jin, Wei Zhao

**Affiliations:** 1State Key Laboratory of Medicinal Chemical Biology, College of Pharmacy, Nankai University, 38 Tongyan Road, Jinnan District, Tianjin 300350, China; fengkekkk@163.com (K.F.); 15706469662@163.com (K.W.); 2120201171@mail.nankai.edu.cn (Y.Z.); hy15031074889@163.com (H.X.); wzhao@nankai.edu.cn (W.Z.); 2Department of Stomatology, Tianjin Nankai Hospital, 6 Changjiang Road, Nankai District, Tianjin 300100, China

**Keywords:** LMWHs, non-anticoagulant activities, clinical applications

## Abstract

Low molecular weight heparins (LMWHs) are derived from heparin through chemical or enzymatic cleavage with an average molecular weight (Mw) of 2000–8000 Da. They exhibit more selective activities and advantages over heparin, causing fewer side effects, such as bleeding and heparin-induced thrombocytopenia. Due to different preparation methods, LMWHs have diverse structures and extensive biological activities. In this review, we describe the basic preparation methods in this field and compare the main principles and advantages of these specific methods in detail. Importantly, we focus on the non-anticoagulant pharmacological effects of LMWHs and their conjugates, such as preventing glycocalyx shedding, anti-inflammatory, antiviral infection, anti-fibrosis, inhibiting angiogenesis, inhibiting cell adhesion and improving endothelial function. LMWHs are effective in various diseases at the animal level, including cancer, some viral diseases, fibrotic diseases, and obstetric diseases. Finally, we briefly summarize their usage and potential applications in the clinic to promote the development and utilization of LMWHs.

## 1. Introduction

In 1916, William Henry Howell and Jay Mcclean discovered that liver extracts prevented blood clotting while testing the validity of procoagulant substances in vivo [1]. At that time, the anticoagulant was thought to be found only in the liver, hence the name heparin [2]. Large-scale production of heparin began in 1937 when Canadian scientists isolated and purified heparin from lung tissue extracts of animals [2]. Although heparin is widely used as an anticoagulant, it is prone to the risk of heparin-induced thrombocytopenia, as well as adverse effects such as osteoporosis and eosinophilia, and requires monitoring of coagulation indicators, making it inconvenient for clinical application. LMWHs, which are obtained by chemical or enzymatic degradation of heparin, have multiple advantages because they have a longer half-life, require less frequent monitoring to check their effectiveness and side effects, and can be administered outside the hospital [3]. For example, Apostolos et al. found that LMWHs were associated with fewer bleeding complications in patients with mild to moderately severe injuries when examining the difference between heparin and LMWHs [4].

In approximately 1980, LMWHs were developed and defined as heparin salts with an average Mw of 6000 Da and at least 60% of chains having a Mw of less than 8000 Da [5]. LMWHs are degraded from heparin; as a result, they have the same monosaccharide composition, polysaccharide chain modification, and oligosaccharide sequences [6]. The chemical structure of LMWH components will be greatly affected by its degradation process. Therefore, different degradation processes have different cutting sites and will also produce different reducing ends (RE) and nonreducing ends (NRE) [7]. There are eight officially approved LMWHs: bemiparin, certoparin, dalteparin, enoxaparin, nadroparin, parnaparin, reviparin, and tinzaparin. The treatment exchange between them is considered inappropriate [8].

LMWHs were initially known for their anticoagulant effect. However, with extensive and continuous research, they have shown good effects on some diseases at the experimental level, such as viral and inflammatory diseases, cancer, obstetrics, and gynecology. In fact, LMWHs have been investigated clinically in COVID-19, but the use of them have not been officially approved. In this review, we describe the preparation methods of LMWHs and highlight the latest clinical studies on their non-anticoagulant activity to guide future research.

## 2. The Structure of LMWHs Varies Depending on the Preparation Process

### 2.1. Nitrite Degradation

Nitrite degradation uses nitrite to split the glycosidic bond of heparin and then uses sodium borohydride to obtain new oligosaccharides by a reductive reaction, and the final product is recovered by methanol precipitation. Experimental conditions, including reaction time, temperature, and pH of the reaction solution, will affect the molecular weight distribution, biological activity, and other physicochemical properties of LMWHs [9]. In the first step, the amount of acid and reaction time is crucial for controlling the chain length and distribution of new oligosaccharides, while in the third step, the concentration of methanol is the key factor in determining whether the oligosaccharides are recovered or discarded according to the size. These two steps determine the Mw of LMWHs. In the second step, the newly formed active aldehydes are converted into alcohol to protect the new oligosaccharides from other side reactions [10]. Therefore, this step does not affect the Mw of the final product. At present, LMWHs prepared by this method contain nadroparin calcium and dalteparin sodium with an Mw of 5600–6400 Da (Figure 1A).

### 2.2. Chemical β-Elimination

Chemical β-elimination cleavage occurs under alkaline conditions. The glycosidic bond breaks, and the reducing end C6 dehydrates with C1 to form a 1,6-inner ether structure. The commercially available LMWH prepared by this method is enoxaparin sodium, with an Mw of 3800–5000 Da (Figure 1B).

### 2.3. Enzymatic β-Elimination

The enzymatic β-elimination method uses specific digestion sites of different enzymes to split heparin chains. Among them, heparinase III does not destroy the core pentasaccharide structure of heparin, so the obtained product has higher anticoagulant activity [11]. Compared with chemical methods, the preparation of LMWHs by enzymatic degradation is milder, and LMWHs with different Mw can be prepared by adjusting the enzymatic hydrolysis time. The Mw of tinzaparin sodium prepared by this method ranges from 5500–7500 Da (Figure 1C).

### 2.4. Peroxide Degradation

Free radical oxidation is also called hydrogen peroxide degradation. This method catalyzes hydrogen peroxide to produce free radicals, degrades the heparin chain, and forms LMWHs through external conditions. The characteristic of this preparation method is that it can randomly break the glycosidic bond and generate odd or even chains [12,13]. At present, commercially available LMWHs prepared by this method are mainly parnaparin sodium with an Mw of 4000–6000 Da (Figure 1D).

### 2.5. Novel Methods for Heparin Depolymerization

In recent years, scientists have improved preparation methods based on these four basic ones. For example, Kyohei et al. used a photocatalytic preparation method with titanium dioxide as the catalyst to obtain highly pure LMWHs [14]. The photochemical activation of a titanium dioxide aqueous solution can produce reactive oxygen species, which can randomly oxidize the sugar residues of polysaccharides, resulting in a decrease in the Mw of the polysaccharide. In addition, the use of ultrasound provides a promising method for depolymerizing heparin. Zhi et al. used an ultrasound-assisted Fenton system to degrade unfractionated heparin synergistically into LMWHs within twenty minutes, with higher anticoagulant activity [15]. In addition to these degradation methods, some researchers have also proposed ideas for chemically enzymatic synthesis of these sulfonated oligosaccharide compounds, which have been realized. It typically simulates the biosynthesis of heparin, starting with the bacteria Escherichia coli K5 polysaccharide heparosan. After chemical and heparinase treatment, low molecular weight n-sulfo, n-acetyl heparosan was obtained, with a target Mw of 4000 Da. Finally, the target product was obtained by o-sulfonation modification [16]. In summary, these novel methods can greatly improve the efficiency of preparation and, more importantly, reduce postprocessing steps, thereby obtaining LMWHs with higher purity and reducing the potential drug risks caused by impure raw heparins. The following summarizes the representative methods for preparing LMWHs in recent years and compares their characteristics in detail (Table 1).

## 3. Non-Anticoagulant Activities of LMWHs

LMWHs retain the highly sulfonated ionic glycan structures from heparin (Figure 2), which may be the reason for binding to a variety of disease-related proteins, such as selectins, integrins, inflammatory factors, and proteases [19]. Except for anticoagulation, LMWHs have the functions of preventing glycocalyx shedding, anti-inflammatory, antiviral infection, inhibiting angiogenesis, inhibiting cell adhesion, and anti-fibrosis, so they are widely used in cancer, some viral diseases, obstetrics and gynecology, systemic inflammation and others (Figure 2).

### 3.1. Potential Effects of LMWHs in Some Viral and Inflammatory Diseases

Sepsis is a complex clinical syndrome caused by bacterial, viral, or fungal infections, resulting in excess mortality and morbidity worldwide [20]. Typical clinical signs associated with sepsis, including cytokine storm, multiple organ dysfunction syndrome, coagulation disorders, and infectious shock, are demonstrated in critically ill patients with neococcal pneumonia [21]. In most autopsy studies, sepsis was present in the majority of neocoronary pneumonia decedents [22]. Accumulating evidence suggests that LMWHs or heparin therapy can reduce mortality in patients with COVID-19 and has multiple benefits for sufferers. For example, a study from Italy showed that heparin was independently associated with reduced mortality in patients over 59 [23]. In addition to anticoagulation, LMWHs or heparin also have other therapeutic functions related to the clinical manifestations of COVID-19, neutralizing inflammatory chemokines, cytokines, and extracellular cytotoxic histones and disturbing leukocyte transport [24]. Other experiments have also shown that it can inhibit the inflammatory response of the glomerulus [25]. Due to their excellent anti-inflammatory effects, patients with sepsis can achieve better survival benefits and coagulation parameters after using these drugs. This may be achieved by restoring the protective proteoglycans on the endothelial surface and down-regulating the levels of serum inflammatory factors such as IL-6 and TNF-α [26].

Therefore, the potential beneficial non-anticoagulant mechanisms of heparin/LMWHs in the treatment of sufferers deserve further investigation.

#### 3.1.1. Preventing Glycocalyx Shedding

The endothelial glycocalyx covers the top membrane surface of vascular endothelial cells and is composed of proteoglycans, glycosaminoglycan chains, and glycoproteins [27]. Heparan sulfate proteoglycans, such as syndecan and glypican, are the main components of the glycocalyx, ranging from 50% to 90% [28]. It can regulate vascular tension and permeability, prevent thrombosis, and affect leukocyte adhesion and inflammatory reactions. Under normal physiological conditions, the endothelial glycocalyx prevents protein leakage in a charge and size-dependent manner [29]. Heparanase is the only mammalian enzyme known to degrade heparan sulfate. Therefore, increasing the activity of this enzyme impairs the glycocalyx, leading to a subsequent loss of endothelial barrier function and vascular leakage, leading to lung and kidney complications [29,30]. For example, Simone et al. found that compared with the healthy control group, the plasma heparanase activity of COVID-19 patients significantly increased [31], which would degrade the endothelial glycocalyx, leading to a variety of complications. At present, heparin/LMWHs and chemical heparin derivatives have been proven to be effective heparanase inhibitors [32]. Clinical data showed that non-ICU patients receiving a prophylactic dose of dalteparin significantly reduced heparanase activity and inhibited glycocalyx disturbance [33]. Other studies have shown that plasma from COVID-19 patients can promote glycocalyx shedding in healthy endothelial cells, which may be related to high levels of cytokines and redox imbalances, while LMWHs may prevent glycocalyx destruction through this approach [31].

#### 3.1.2. Anti-Inflammatory

Novel coronavirus pneumonia is associated with high levels of proinflammatory cytokines [34]. Cytokine storm has been proven to be one of the important reasons for the deterioration of people with COVID-19 [35]. Several studies have found that serum cytokines in patients with COVID-19 are significantly increased, including IL-1, IL-6, IL-7, IL-8, IL-9, IL-10, granulocyte colony-stimulating factor (G-CSF), granulocyte-macrophage colony-stimulating factor (GMCSF), chemokines (CXCL10, CCL2, CCL3), IFN-γ, TNF-α and vascular endothelial growth factor (VEGF) [36,37,38]. IL-6 released by activated macrophages is the key to initiating a cytokine storm [39]. The production of these cytokines can lead to severe symptoms of novel coronavirus, including acute respiratory distress syndrome (ARDS), hypotension, and disseminated intravascular coagulation [36,37,40]. Research shows that LMWHs can interact with chemokines and cytokines, which are produced in the “cytokine storm” of COVID-19, and inhibit the release of different cytokines (IL-4, IL-5, IL-13, and TNF-α) [41].

In addition, heparanase and endothelial glycocalyx are also associated with some inflammatory processes. The increase in heparanase activity is involved in the development of the proinflammatory glycocalyx. Heparan sulfate fragments, caused by heparanase, contribute to inflammation of the extracellular environment. Under these conditions, the structure of the glycocalyx changes, promoting the binding of chemokines, selectin, and integrin expressed by leukocytes [30,42]. Heparin/LMWHs can combine with leukocyte surface ligands (selectin and integrin) to interfere with leukocyte rolling, adhesion, and migration [43], that is, to prevent leukocytes from gathering and transporting to inflammatory sites, thereby inhibiting the immune response and weakening the damage of viruses to the human body. The glycosaminoglycan on the surface of endothelial cells can regulate the bradykinin pathway; specifically, degraded heparin promotes the hydrolysis of kininogen to produce bradykinin [44]. Therefore, low molecular weight heparin may inhibit the formation of slow kinin by inhibiting heparanase activity and binding to high molecular weight kininogen, thereby reducing local inflammation and vascular leakage of COVID-19 [45]. Finally, the role of heparin/LMWHs may also be related to the inhibition of complement activation [46].

Heparin/LMWHs may promote the inhibition of inflammatory responses through various mechanisms. Because of this, LMWHs can be used not only in COVID-19 infections but also in other inflammatory diseases, such as experimental models of bronchial asthma, ulcerative colitis, burns, ischemia-reperfusion, arthritis, and peritonitis [44], with good anti-inflammatory effects.

#### 3.1.3. Reducing Virus Entry into Cells

Endothelial cells express angiotensin-converting enzyme 2 (ACE2), which is the entrance of novel coronavirus (SARS-CoV-2). In the process of studying the structure of SARS-CoV-2, a new glycosaminoglycan binding motif in spike glycoprotein (SGP) was found, which is responsible for the binding and fusion of ACE2 of the virus and host membrane [47]. These co-receptors on the cell membrane lead to an increased concentration of virus around the host cell and maximize infection [48]. The plaque inhibition test of Vero E6 cells showed that heparin interacted with the spike protein of SARS-CoV-2 to prevent the virus from binding to the host cell receptor ACE2 [49], indicating that heparin/LMWHs have resistance to SARS-CoV-2. On the other hand, it was found that they may inhibit viral infection by inhibiting the main proteinase (Mpro, also known as nonstructural protein 5) of SARS-CoV-2 [44]. As a key enzyme of coronavirus, Mpro plays a crucial role in mediating viral replication and transcription, making it an attractive drug target for SARS-CoV-2 [50]. Li et al. found that the IC_50_ of heparin on Mpro was 7.8 ± 2.6 nM, showing strong inhibition of the Mpro activity of SARS-CoV-2 [51]. Molecular docking results show that sulfate groups can form hydrogen bonds with Cys145 residues at the active site of Mpro [52], which may be the reason why heparin containing multiple sulfate groups combines with Mpro. Based on the above research, LMWHs are likely to be developed into new antiviral drugs in the future.

### 3.2. Potential Effects of LMWHs in Cancer

#### 3.2.1. Inhibiting Angiogenesis

The formation of new blood vessels is necessary for tumor cell growth and metastasis. It provides the tumor with oxygen and nutrients, and it is also the medium of tumor cell metastasis. Fibroblast growth factors (FGFs) are the first recognized angiogenic factors, and their roles in endothelial cell proliferation, migration, adhesion, and other angiogenesis processes have been widely recognized [53]. Fibroblast growth factor 2 (FGF2) can induce the expression of VEGF in vascular endothelial cells. Fibroblast growth factor receptor 1 (FGFR1) and vascular endothelial growth factor receptor (VEGFR) are the main receptors for promoting angiogenesis. Its angiogenic effect may be based on the correlation between FGF-VEGF signaling pathways. Through the synergistic effect of these pathways, tumor angiogenesis and growth will eventually be amplified [54]. An experiment investigated the effects of heparin on FGF and blood vessels in mice, and the results showed that the tumors were smaller and angiogenesis was reduced in the heparin-treated group. This explains the possible mechanism: heparin isolates FGF and inhibits its binding to receptors; that is, heparin may destroy the interaction between FGF and heparin proteoglycan [55] (Figure 3). Similar results were obtained when using a variety of LMWHs (dalteparin, enoxaparin, and dinzaparin), of which dalteparin was the most effective drug. Similar to a previous study, the authors concluded that LMWHs can reduce angiogenesis and subsequent tumor growth by isolating FGFs from low-affinity receptors on tumor cells [56].

#### 3.2.2. Inhibiting Cell Adhesion

The term “cell adhesion” refers to the broad field of cell-cell and cell-substrate adhesion. Cells attach to adjacent ones by direct or indirect contact with molecules on their surfaces and maintain the organizational framework together with the extracellular matrix [57]. Cell adhesion occurs when adhesion molecules interact with trans-membrane proteins on the cell surface. This effect is comprehensive and not just a bio-mechanical process that “glues” cells together [57]. Through this adhesion, signals can be transmitted between cells to stimulate the cell cycle, cell differentiation, cell migration, and cell survival regulation [58]. Metastasis and the spread of cancer cells are difficult problems in cancer treatment. Most cancer deaths are caused by the spread of cancer cells, which invade the tight micro-environment from the primary site, enter the vascular system, and spread to the permitted remote organs to form new tumors. Some articles have pointed out that adhesion molecules play a decisive role in enhancing the recurrence, invasion, and distant metastasis of cancer [59].

Clinical studies have shown that treatment with LMWHs can significantly increase the survival time of the sick, primarily through its inhibition of cell-cell interactions [60]. Fraxiparine inhibited the adhesion, invasion, and migration of A549 cells [61]. Another study investigated whether LMWHs could significantly inhibit the adhesion and migration of PC-3M cells on endothelial cells. There are several possible reasons. First, the anti-metastasis effect of LMWHs is to inhibit the selectin [62] (Figure 3). Selectin is an adhesion molecule related to immune function [63]. Many studies have demonstrated that it is involved in pathophysiological processes, including cancer metastasis [64]. Among them, P-selectin promotes the interaction between platelets and mucins in tumor cells [65,66], which is considered to be an important aspect of hematogenous metastasis. In addition, it has been reported that E-selectin enhances tumor cell adhesion, metastasis, and diffusion [67,68], and L-selectin can enhance tumor cell exosmosis by mediating leukocyte aggregation. An experiment with mice of P- and L-selectin knockout genes showed that the interaction between platelets and tumor cells was reduced, thus reducing the metastasis of tumor cells [69]. Second, Bendas et al. believed that LMWHs achieve anti-metastasis functions by interfering with the very late anti-gen-4/vascular cell adhesion molecule-1 (VLA-4/VCAM-1) pathway [70] (Figure 3). Under physiological conditions, integrin VLA-4 is expressed in various types of leukocytes and tumor cells. VLA-4 can bind to the activated endothelial expressed ligand VCAM-1, which can mediate the adhesion and metastasis of tumor cells [70]. Finally, cadherin is a calcium-based transmembrane protein that plays an indispensable role in cell adhesion, maintaining cell morphology and tissue structure. Tinzaparin is involved in up-regulating E-cadherin expression in malignant cells in a mouse model of pancreatic cancer, and down-regulating this expression contributes to promoting metastasis through local invasion and migration [71] (Figure 3). The studies above may confirm another possible mechanism of the anti-metastasis effect. Many clinical data have shown that LMWHs have a certain inhibitory effect on cancer. However, when conducting research, the type of LWMHs should be considered because different preparations have dissimilar effects on cell adhesion and metastasis.

### 3.3. Potential Effects of LMWHs in Fibrotic Diseases

Fibrosis is a pathological change that deviates from the physiological wound-healing response. Its formation is a dynamic and gradual process involving complex cellular and molecular mechanisms related to inflammation and can occur in almost any organ, such as the heart, lungs, liver, skin, and kidneys [72]. Usually caused by infection, local ischemia, inflammation, autoimmune diseases, etc., fibrosis begins with frequent or sustained triggering of parenchymal cell damage or death, and its repair process cannot restore balance. The persistence of inflammation and changes in immune cell composition are typical features of prefibrosis [73]. These immune cells promote endothelial cell fibrosis by secreting fibroblast growth factors and promote tissue damage by stimulating angiogenesis and the secretion of matrix metalloproteinases (MMP) and reactive oxygen species (ROS) [74,75,76]. TGF-β, secreted by parenchymal cells and white blood cells, is the main regulatory factor for fibrosis. In addition, Th17 and Th2 cells can secrete IL-17, IL-4, and IL-13, which can activate fibroblasts and stimulate the production of ECM proteins [77]. Excessive ECM deposition hinders oxygen diffusion, leading to tissue hypoxia. This situation exacerbates the vicious cycle of fibrosis throughout the entire organ.

LMWHs have anti-fibrotic effects by regulating the secretion of various cytokines and growth factors [78]. They can also be used in conjunction with other drugs to prevent or treat fibrotic diseases in clinical practice (Figure 4). In clinical practice, LMWHs can be used as anti-fibrotic drugs for patients with chronic hepatitis B [79]. According to Takashi et al., LMWH treatment can promote the production of hepatocyte growth factor (HGF), which may inhibit the production of TGF-β1, thereby reducing the progression of liver fibrosis [80]. Other mechanisms suggest that HGF can promote the generation of MMP-1, a type of enzyme that can digest fibrotic collagen, leading to a decrease in fibrosis [81]. Mesenchymal stem cells (MSCs) can be transformed into a variety of mesenchymal cells, which can secrete HGF, and thus, they have anti-inflammatory and anti-fibrotic effects. Takashi et al. cocultured LMWHs and adipose-derived mesenchymal stem cells (ASCs) in culture medium and then injected them into interstitial lung disease model mice. LMWHs significantly increased the number of mASCs and enhanced their migratory, anti-inflammatory, and anti-fibrotic effects compared to controls [78].

Moreover, Li et al. found that long-term treatment with LMWHs for CCl4-induced fibrosis led to a decrease in the number of activated hepatic stellate cells (HSCs) [82], which are directly involved in the pathological changes in liver fibrosis [83]. Western blotting and EMSA experiments showed that LMWH administration resulted in a decrease in the levels of collagen types I and III in the livers of rats from the model group. In addition, in vitro studies on HSCs showed that LMWHs had an inhibitory effect on ERK and phosphorylation levels and reduced the activity of activator protein-1 (AP-1, the switch that initiates gene transcription and regulates cell growth and differentiation), thus reducing HSC proliferation and avoiding further development of fibrosis [82]. In a dimethylnitrosamine-induced liver fibrosis model, Lee et al. found that heparin–pluronic nanogel inhibited the mRNA expression of type I collagen and MMP-1 and MMP-2 tissue inhibitor, and it reduced important molecules of the TGF-β/Smad signaling pathway, such as TGF-β1, p-Smad 2 and p-Smad 3 [84]. In conclusion, heparin–pluronic nanogel displays an anti-fibrotic effect by inhibiting the TGF-β/Smad pathway as well as eliminating the extracellular matrix.

### 3.4. Potential Effects of LMWHs in Obstetrical Diseases

In the last century, antiphospholipid syndrome (APS) was found to be associated with recurrent pregnancy loss, which can lead to arterial or venous thrombosis repeatedly and pregnancy diseases related to antiphospholipid antibodies [85]. Obstetric APS, also known as a major obstetric syndrome, has a high incidence and can be detrimental to both the mother and fetus throughout pregnancy. Its effects include recurrent miscarriage of premature infants, ischemic placental insufficiency, and late pregnancy complications, such as pre-eclampsia, eclampsia, stillbirth, intrauterine growth restriction (IUGR), placental abruption, and premature delivery [86,87]. One of the conditions for a successful pregnancy is proper uteroplacental circulation [88]. Previous studies have shown that placental function and development defects may lead to pregnancy loss, which may be caused by arterial thrombosis [89]. In addition, venous thromboembolism (VTE) is more common in pregnancy than arterial thrombosis [90]. The blood in the fetus starts to flow in the villous space of the placenta around the 10th week of pregnancy, which can ensure the transportation of nutrients from the maternal blood to the fetal tissue [91]. The tendency of thrombosis in pregnancy can lead to pregnancy loss, placental damage, and fetal death. Farquharson et al. found that APS could increase thrombin production [92], so antithrombotic and anticoagulant therapy could be applied to such diseases [93]. Clinically, dalteparin and enoxaparin are often used to treat APS [94]. Monica et al. found that the use of LMWHs can reduce the incidence of pre-eclampsia and other placental complications when treatment begins before 16 weeks of pregnancy in high-risk women [95]. Compared with aspirin alone, their combination can significantly reduce the risk of pre-eclampsia. Kelsey et al. studied the cardiovascular effect of LMWHs on high-risk pregnant women with pre-eclampsia and found that the angiogenesis of endothelial cells in the serum of high-risk women was impaired, and the transcription of placental growth factor-1 and placental growth factor-2 was increased [96]. LMWHs can improve angiogenesis and promote the secretion of placental growth factors. Therefore, LMWHs may improve maternal endothelial function in high-risk pre-eclampsia pregnant women, but the specific mechanism remains to be studied.

### 3.5. Potential Effects of LMWHs in Other Diseases

LMWHs have many action sites and diverse action modes in vivo, so the types of diseases that can be treated with LMWHs are also constantly explored. LMWHs (such as enoxaparin) can significantly reduce the severity of proteinuria in patients with diabetes nephropathy, reduce proteinuria, glycosaminoglycans, and the renal cycle, and promote clinical remission in patients with hormone-sensitive nephrotic syndrome [97]. The reason may be that they inhibited heparanase, thus avoiding the degradation of glycosaminoglycan sulfate in the glomerular basement membrane and preventing blood protein from leaking into urine [98]. In addition, in a rat model of diabetes nephropathy, modified heparin inhibited TGF-β1 overexpression to prevent diabetes glomerulosclerosis [99].

## 4. Various Heparin Conjugates with Non-Coagulant Effects

Heparin/LMWH conjugates have been developed from heparin to retain their pharmacological effects while reducing its shortcomings, e.g., the necessity of injectable dosage and the anticoagulant activity that may predispose patients to bleeding. For example, a new heparin-derived non-anticoagulant angiogenesis inhibitor, LMWH-taurocholate conjugate, was reported by Seung Woo Chung et al. [100]. This chemical modification reduces the binding affinity of LMWHs for antithrombin and blocks angiogenic factors involved in the initial angiogenesis (VEGF and FGF2) and the primitive stabilization of the endothelial vascular network (platelet-derived growth factor subunit B) to inhibit multiple stages of angiogenesis, which directly leads to decreased blood perfusion throughout the tumor and inhibition of tumor growth. Due to the high expression of heparinase in tumor tissues and the fact that heparin can be specifically degraded in the tumor region, Zaixiang Fang et al. synthesized heparin–paclitaxel and heparin–pyrophosphopyridine and then co-assembled the two couplings into a nanomedicine [101]. The nanoparticles increased the accumulation of pyrophosphopyridine by 11-fold in tumors and inhibited the tumor growth of 4T1-loaded tumor-bearing mice by up to 98.1%, with no systemic toxicity. In addition, these conjugates may inhibit tumor growth through multiple mechanisms. The heparin–taurocholate conjugate not only reduces the proliferation of cancer cells but also reduces the production of vascular endothelial growth factor through ERK dephosphorylation [102]. By dephosphorylating VEGFR, ERK, and FAK proteins, heparin-bovine cholate conjugate effectively reduced endothelial cell migration, invasion, and tube formation. Jae Hyeon Lee et al. prepared nanoparticles by coupling undisolated heparin with doxorubicin [103]. In a BALB/c animal model containing ct26, doxorubicin coupled with heparin, which produced ROS by cytotoxicity, exhibited inhibitory effects on tumor growth and metastasis. Jeong Uk Choi et al. synthesized LMWH-bile acid conjugates (LMWH with four dimeric dextrocholic acid molecules), which selectively target the tumor endothelial cell markers, doppel, and the doppel/VEGFR2 axis and capture VEGF, with the trio forming an intermediate complex that further inhibits the process of tumor angiogenesis [104].

To improve the drug dosage form, Hae Hyun Hwang et al. developed an orally available heparin by covalently bonding lactoferrin to heparin [105]. In this, lactoferrin can be absorbed orally and interacts with lactoferrin receptors expressed on the gut, blood-brain barrier, and glioma tumor mass. The conjugate is specifically delivered to the brain tumor via receptor cytophagy and attenuates angiogenesis by reducing the activity of growth factors around the tissue. In addition, this heparin coupling can also be applied to renal diseases. Abhishek Sahu et al. coupled heme with heparin and self-assembled it to form nanoparticles [106]. After selectively injecting this into the injured kidney via intravenous administration, it effectively scavenges ROS, reduces inflammation, and shows good therapeutic effects in acute kidney injury models. In the nervous system, heparin/substance p/PLCL coupling can inhibit cell adhesion, improve the growth and development of nerve cells in vitro, inhibit the inflammatory response of activated microglia and reactive astrocytes, reduce the formation of glial sheaths, and promote the regeneration of nerve cells [107].

Insulin-like growth factor-1 receptor (IGF-1R) is a receptor tyrosine kinase that plays a key role in abnormal cell growth, tumor invasion, and metastasis and is involved in the development of breast, colorectal, lung, and liver cancers. Short-chain oligosaccharides (HS06) can enter a unique pocket in the IGF-1R ectodomain and compete with the natural cognate ligand IGF-1, thereby blocking the physiological processes mediated by IGF-1R and exerting anti-cancer effects [108]. Additional studies have shown that HS06 activates p38α/β mitogen-activated protein kinase, followed by inhibition of TCF4 signaling, ultimately inducing apoptosis in cancer stem cells [109]. Pixatimod (PG545: Figure 5) contains oligosaccharides and cholesterol fragments, which not only inhibit the enzyme acetylheparanase but also give the drug a potent anti-cancer effect due to the lipophilic fraction [110]. It targets tumor-associated macrophages via the inhibition of heparanase and can also activate NK cells [110]. In particular, when pixatimod is combined with an anti-PD-1 antibody, the rate of tumor inhibition is two times higher than that with PD-1 alone [111].

In summary, as research on LMWHs deepens, researchers have discovered activities beyond anticoagulation, as mentioned earlier. In clinical practice, it is mainly used for deep vein thrombosis (42%), which forms in the deep vein of the leg or pelvis, cancer (22%), peritoneal disease (10%), abortion and pregnancy (9%), and renal insufficiency (8%) [112]. The data also show that the top five types used in clinical trials are dalteparin, enoxaparin, tinzaparin, nadroparin, and bemiparin [112]. At present, many LMWHs and their analogs are in clinical trials, including ODSH in pediatric cancer patients (NCT02164097); Tinzaparin in cancer (NCT00475098) [113]; PG545 in solid tumors (NCT02042781) [111]. Table 2 summarizes some clinical application examples to learn more. We can see that although the diseases treated with LMWHs vary, they mainly rely on anticoagulation and inhibition of inflammatory factors to exert their effects. The table also demonstrates the combined use of some drugs and their therapeutic effects on diseases.

## 5. Conclusions and Prospects

As an anticoagulant and antithrombotic drug, heparin has been active in the clinic for more than 80 years. LMWHs, derived from unfractionated heparin, have several advantages, including higher bioavailability, longer half-life, lower dose and longer action time, and less binding with other proteins to reduce its serious side effects. LMWHs is a general term for a variety of products whose preparation methods result in slightly different structures, so they are generally not used interchangeably. With continuous research, novel preparation methods have emerged, such as photocatalysis and ultrasonic degradation, which solve the problems of introducing redundant chemicals, difficult separation, and low yield. In particular, enzymatic chemical synthesis makes the structure of LMWHs more accurate and eliminates some impurities in the cracking method. More importantly, enzymatic chemical synthesis has potentially solved the problem of shortage of heparin materials, mainly from pig intestinal mucosa. Although these new methods have many advantages, there is still a long way to go before they can be applied in production and clinical practice because they require more detailed data on the new LMWHs, including their biological activity, pharmacological efficacy, and safety in vivo. Moreover, the complex mechanisms in cancer, obstetrics, and other diseases also need to be further clarified. The dosage, mode of administration, and how to select the appropriate LMWHs need further research to ensure the safety and effectiveness of the medication.

Currently, with the continuous development of drug delivery systems, it may be a future trend to develop new dosage forms for LMWHs. LMWHs are generally used for injection in clinical practice because of their high anion charge density, low intestinal absorption rate [126], first-pass effect, and low oral bio-availability. It adds many inconveniences for ordinary people to use it at home alone. Second, LMWHs can combine with a variety of proteins, which may cause side effects when used. Therefore, the study of new LMWH dosage forms and drug delivery systems is very important for the further application of LMWHs. For example, Ozemre et al. explored nanodelivery systems for enoxaparin to avoid the influence of the gastrointestinal tract, making oral administration possible [127]. Some scientists have prepared LMWHs in a nasal inhalation form. Clinical studies have shown that it can prevent SARS-CoV-2 from adhering to nasal epithelial cells, effectively preventing infection by the virus [128]. In addition, this dosage form of LMWHs is easy to apply and carry and has potential use for possible lung diseases such as acute lung injury, chronic obstructive pulmonary disease, cystic fibrosis, and asthma [129].

Finally, new forms of LMWHs have better compliance, fewer adverse effects, and more precise targeting. With enough experimental data, LMWHs can enter a new era of medical treatment.

## Figures and Tables

**Figure 1 pharmaceuticals-16-01254-f001:**
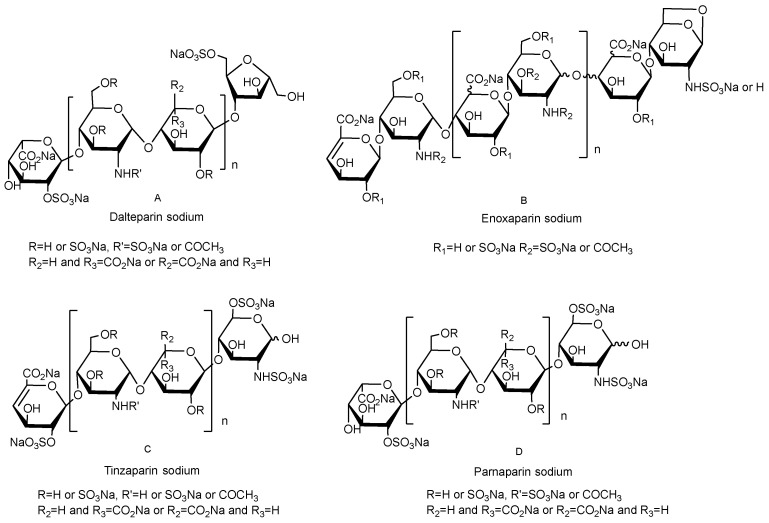
The structures of dalteparin sodium (**A**), enoxaparin sodium (**B**), tinzaparin sodium (**C**) and parnaparin sodium (**D**).

**Figure 2 pharmaceuticals-16-01254-f002:**
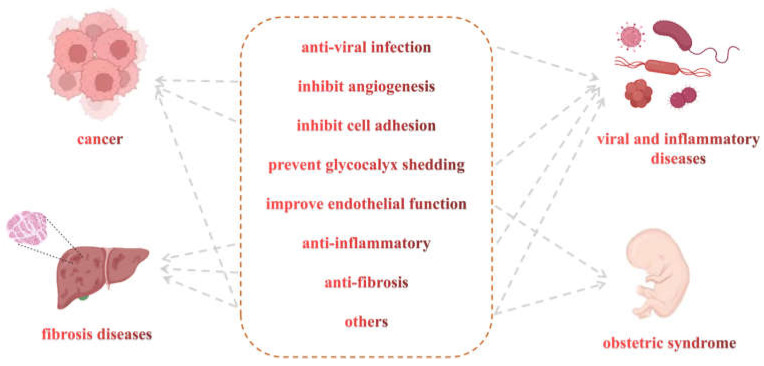
The relationship between the potential mechanism of LWMHs and diseases.

**Figure 3 pharmaceuticals-16-01254-f003:**
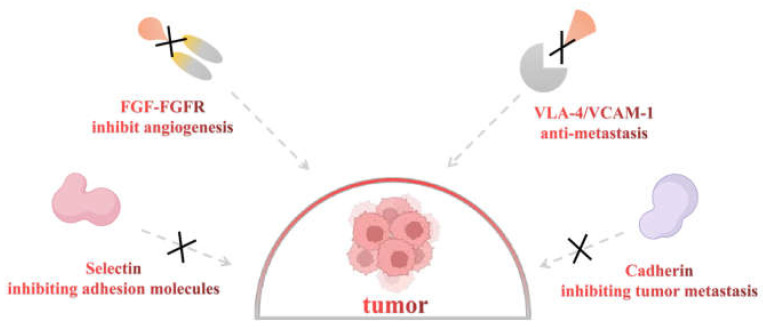
The potential roles of LMWHs in cancer. (1) Inhibiting the interaction between FGF-FGFR to hinder angiogenesis; (2) inhibiting selectin, up-regulating of cadherin expression in malignant cells and interfering VLA-4/VCAM-1 pathway to hinder cell adhesion and metastasis.

**Figure 4 pharmaceuticals-16-01254-f004:**
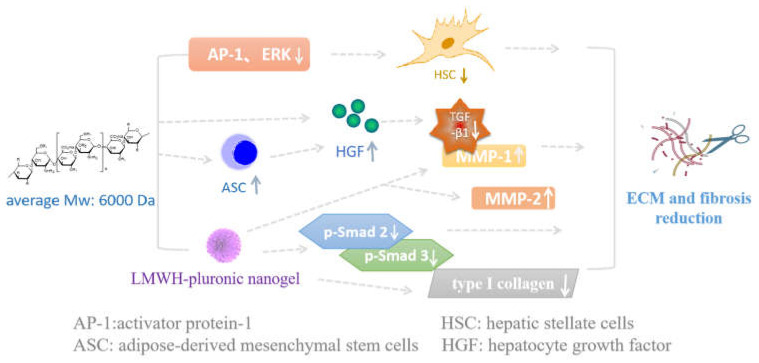
Possible mechanisms of LMWHs inhibiting fibrosis. There are many ways that LMWHs can affect fibrosis: (1) Increase the number of ASCs and secrete more HGF. HGF may inhibit TGF-β and promote the production of MMP-1, thus avoiding fibrosis; (2) reduce ERK, phosphorylation levels, and AP-1, then reduce HSC and avoiding fibrosis; (3) suppress TGF-β/Smad pathways and elimination of extracellular matrix exhibit anti-fibrosis effects.

**Figure 5 pharmaceuticals-16-01254-f005:**
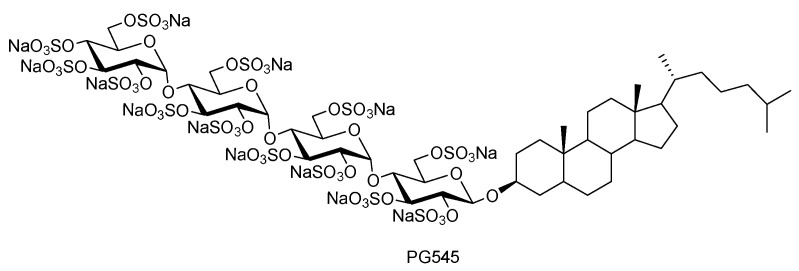
The structure of PG545.

**Table 1 pharmaceuticals-16-01254-t001:** Novel preparation methods of LMWHs and their characteristics.

Preparation Methods	Essence	Advantages	Mw/Da	Ref.
Preparation from bovine and ovine heparins	nitrous acid cleavage	expanding the animal source of heparin preparation	average: 5971 (bovine), 6131 (ovine)	[9]
Enzymatic Preparation with photoswitchable heparinase III	enzymatic β-elimination	artificially controlled enzymatic degradation; well-distributed molecular weight; high anticoagulant activity		[11]
Preparation of a new NG-LMWH	chemical β-elimination	the same activity as heparin in anti-FXa and anti-FIIa; high safety by neutralizing protamine	5400~9000	[17]
Ultrasonic-assisted preparation	free radical oxidation	fewer chemical impurities; high anticoagulant activity	~6300	[18]
Preparation from the ultrasound-assisted Fenton system	free radical oxidation	efficient and convenient; high anticoagulant activity	~4870	[15]
Enzymatic synthesis from n-sulfo heparosan depolymerized by heparinase or heparin lyase	synthesis	not dependent on animal sources; the same activity as enoxaparin in Anti-FXa and anti-FIIa; high purity	~4000	[16]

**Table 2 pharmaceuticals-16-01254-t002:** Examples of the clinical application of LMWHs.

Medicines	Applicable Disease	Mechanisms	Results	Ref.
LMWH and Ulinastatin	acute pancreatitis in children	improve the coagulation function	enhanced the efficacy of conventional treatment and reduced mortality	[114]
LMWH and Magnesium sulfate	severe pre-eclampsia	reduce the infiltration ability of cytotrophoblasts and regulate endothelial cell function	improved neonatal survival rate	[115]
enoxaparin	prevention of venous thromboembolism in COVID-19		reduced mortality, clinical deterioration, and venous thromboembolism	[116]
LMWH and Dexamethasone	partial splenic embolization in cirrhotic patients with massive splenomegaly	improve the coagulation function	reduced the incidence of complications (such as post-embolization syndrome, portal vein thrombosis, refractory ascites)	[117]
LMWH calcium and Xueshuantong injection	elderly acute deep venous thrombosis patients	improve the coagulation function and inhibit thrombosis	effectively treated acute deep venous thrombosis patients	[118]
enoxaparin	active ulcerative colitis	anti-inflammatory	significantly improved ulcerative colitis	[119]
LMWH calcium	henoch–schonlein purpura nephritis	inhibit high fibrinolysis to exert anticoagulant effects	showed good curative effect on proteinuria, alleviated the renal injury and protected the kidney function	[120]
LMWH and Amikacin Sulfate	severe senile pneumonia	improve the coagulation function and inhibit pathogen adhesion	promoted the recovery of patients	[121]
LMWH	acute exacerbation of chronic obstructive pulmonary disease	improve the coagulation function and anti-inflammatory	significantly reduced the risk of thrombosis	[122]
nadroparin calcium	epithelial cell ovarian carcinoma	increase HGF serum concentration	enhancers for ovarian cancer chemotherapy drugs	[123]
LMWH	sepsis	improve inflammation and coagulation function	significantly reduced the 28-day mortality rate in patients with sepsis	[124]
LMWH	liver fibrosis in patients with chronic hepatitis B	inhibit collagen proliferation in liver tissues	inhibited liver fibrosis and used as anti-fibrosis drug	[79]
tinzaparin	localized lung cancer		no significant effect on overall survival and recurrence-free survival	[113]
Cytarabine and Idarubicin with heparin derivative	acute myeloid leukemia	block the CXCL12/CXCR4 axis; inhibits platelet factor-4	made the human body have good tolerance and enhanced the therapeutic effect	[125]

## Data Availability

Not applicable.

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
