# Peer review of "Non-Anticoagulant Activities of Low Molecular Weight Heparins—A Review"

_pharmaceuticals, 2023, doi:10.3390/ph16091254_

Round 1

Reviewer 1 Report

This is a well-documented paper on the manufacture and other effects of heparin. It is recommended to revise the important parts to be modified in the text.

1. There are no major grammatical errors, but English is generally awkward, so English correction by an expert is recommended.

2. It is needed to add a new section on "4. Various heparin conjugates with non-coagulant effect" in the manuscript. Various Heparin conjugates are recently developed for the "non-coagulant effect of LMWH" of the review title.

3. Throughout the text (3.1...) and in Figures 1 and 2, covid-19 is a temporary issue. It would be better to unify and express it as an antiviral effect or something.

4.It is recommended to insert and highlight the anti-inflammatory effect picture in Figure 3.

5."2. Basic preparation methods and latest progress of LMWHs" The title is too broad. And note that this is not a review of manufacturing methods (based on the title).

6. The information on fibrosis(obstetrical diseases) and heparin is interesting because heparin reviews do not cover it well. However, it is difficult to understand clearly with Figure 4. (A little more specific expression is needed.)

7.Correct the format of the references. In particular, journal names

8."Table 1. Recent progress in preparation of LMWHs" It's a method that's been done for a long time, but it's hard to see where it's the latest. 

There are no major grammatical errors, but English is generally awkward, so English correction by an expert is recommended.

Author Response

We would like to thank you for your careful reading, helpful comments, and constructive suggestions, which have significantly improved the presentation of our manuscript. We have carefully considered all comments and revised the manuscript accordingly. In the following section, we summarize our responses to each comment. We hope our revised manuscript can be accepted for publication.

Comments from Reviewer 1:

This is a well-documented paper on the manufacture and other effects of heparin. It is recommended to revise the important parts to be modified in the text.

Point 1: There are no major grammatical errors, but English is generally awkward, so English correction by an expert is recommended.

Response 1: According to the reviewer’s kind suggestion, we have carefully examined the whole manuscript and the lingual deficiencies have been corrected in the revised manuscript.

Point 2: It is needed to add a new section on "4. Various heparin conjugates with non-coagulant effect" in the manuscript. Various Heparin conjugates are recently developed for the "non-coagulant effect of LMWH" of the review title.

Response 2: According to the reviewer’s kind suggestion, we added a new section 4 to describe the applications of the various heparin conjugates with non-coagulant effect. (Lines 387-416, Pages 11-12, words highlighted in yellow).

Point 3: Throughout the text (3.1...) and in Figures 1 and 2, covid-19 is a temporary issue. It would be better to unify and express it as an antiviral effect or something.

Response 3: We agree with the reviewer that it would be more scientific to change covid -19 into viral diseases, as shown in the revised manuscript. (Line 128, 19 and 123, words highlighted in yellow).

Point 4: It is recommended to insert and highlight the anti-inflammatory effect picture in Figure 3.

Response 4: According to the reviewer’s suggestion, we have inserted and highlighted the anti-inflammatory effect picture in the new Figure 2. (Page 4).

Point 5: "2. Basic preparation methods and latest progress of LMWHs" The title is too broad. And note that this is not a review of manufacturing methods (based on the title).

Response 5: Thank you very much for your kind suggestion. The title was changed to “2. The structure of LMWHs varies depending on the preparation process”. (Line 56).

Point 6: The information on fibrosis(obstetrical diseases) and heparin is interesting because heparin reviews do not cover it well. However, it is difficult to understand clearly with Figure 4. (A little more specific expression is needed.).

Response 6: We thank the reviewer for pointing out this important issue. We have revised Figure 4 (Page 10) and added some expressions (Page 9-10, words highlighted in yellow) to better explain the possible applications of heparin in the treatment of fibrotic diseases.

Point 7: Correct the format of the references. In particular, journal names.

Response 7: According to the reviewer’s suggestions, we have uniformed the format of the references in the revised manuscript.

Point 8: "Table 1. Recent progress in preparation of LMWHs" It's a method that's been done for a long time, but it's hard to see where it's the latest.

Response 8: Thank you very much for your kind suggestion. The table caption was changed to “Table 1 Novel preparation methods of LMWHs and their characteristics”. (Line 116, words highlighted in yellow).

Reviewer 2 Report

This short review deals with physiological activities of Low Molecular Weight Heparin (LMWH) preparations other than those associated with anticoagulant activity. The focus on LMWH is welcome, as these products are known to have better bioavailability and  reduced side effects as compared with unfractionated heparin, and have already been through clinical trials.

The review first summarises current methods of preparation of LMWH products, and some that may be used in the future. Then the effects of heparin on viral infections (with a strong emphasis on Covid-19), glycocalyx stability, and inflammation are briefly described before a more detailed discussion of specific conditions such as cancer, fibrosis, plus obstetric and other diseases. A useful table of LMWH clinical applications is provided, demonstrating that LMWH is already giving positive results in patients for several indications other than thrombosis.  

The review is written in a concise style, in reasonably good English. Though it is not comprehensive it provides a good introduction to the subject. A few simple changes are suggested, as follows:

Abstract line 11 is ambiguous: does it mean ‘with an average molecular weight (Mw) in the range 2000-8000 Da’?

Lines 34-35: this sentence is not clear. Does ‘require less blood’ mean ‘require less frequent monitoring’? Also please alter ‘do not have to be administered in the hospital where heparin is required’ to read ‘can be administered outside the hospital’.

In section 2, please start a new numbered paragraph for each preparation method, 2.1 Nitrite degradation; 2.2 Chemical b-elimination; 2.3 Enzymatic b-elimination 2.4 Peroxide degradation, then 2.5 Novel methods for heparin depolymerisaton.

Lines 77 to 78 Please cite the source of the statement ‘heparanase III does not destroy the core pentasaccharide structure of heparin, so the obtained product has higher anticoagulant activity’.

Lines 177, 185 and elsewhere: the term ‘heparin sulfate’ has no clear meaning, please change to either ‘heparan sulfate’ or ‘heparin’.

Line 258: ‘P-selection’ should be ‘P-selectin’; Line 260: ‘E-selection’ should be ‘E-selectin’ Line 261, line 262: ‘L-selection’ should be ‘L-selectin’

Line 386: Please change ‘used instead’ to ‘used interchangeably’.

Lines 390-391: Please add the word ‘potentially’ to read ‘enzymatic chemical synthesis has potentially solved the problem of shortage of heparin materials mainly from pig intestinal mucosa’. At present chemoenzymatically synthesized heparins are not available in sufficient quantities or at low enough cost, and their efficacy and safety have not been demonstrated.

Lines 406-407 Please alter ‘Gamze et al developed a nanometer preparation of enoxaparin’ to read  ‘Ozemre et al. explored nanodelivery systems for enoxaparin’

In the final section on new drug delivery systems it might be good to mention inhalation of nebulized heparin to treat lung conditions including SARS-CoV-2: for example see Eder et al (mBio. 2022 Dec 20;13(6):e0255822. doi: 10.1128/mbio.02558-22) and the recent review by Shute  (Pharmaceuticals 2023, 16(4), 584; https://doi.org/10.3390/ph16040584).

The quality of English is not perfect but it is not difficult to understand. 

Author Response

We would like to thank you for your careful reading, helpful comments, and constructive suggestions, which have significantly improved the presentation of our manuscript. We have carefully considered all comments and revised the manuscript accordingly. In the following section, we summarize our responses to each comment. We hope our revised manuscript can be accepted for publication.

Comments from Reviewer 2:

This short review deals with physiological activities of Low Molecular Weight Heparin (LMWH) preparations other than those associated with anticoagulant activity. The focus on LMWH is welcome, as these products are known to have better bioavailability and  reduced side effects as compared with unfractionated heparin, and have already been through clinical trials.

The review first summarises current methods of preparation of LMWH products, and some that may be used in the future. Then the effects of heparin on viral infections (with a strong emphasis on Covid-19), glycocalyx stability, and inflammation are briefly described before a more detailed discussion of specific conditions such as cancer, fibrosis, plus obstetric and other diseases. A useful table of LMWH clinical applications is provided, demonstrating that LMWH is already giving positive results in patients for several indications other than thrombosis.

The review is written in a concise style, in reasonably good English. Though it is not comprehensive it provides a good introduction to the subject. A few simple changes are suggested, as follows:

Point 1: Abstract line 11 is ambiguous: does it mean ‘with an average molecular weight (Mw) in the range 2000-8000 Da’?

Response 1: Yes, it means ‘with an average molecular weight (Mw) in the range 2000-8000 Da’. Thanks the reviewer’s comments, we have corrected this in the revised manuscript. (Line 10, words highlighted in yellow).

Point 2: Lines 34-35: this sentence is not clear. Does ‘require less blood’ mean ‘require less frequent monitoring’? Also please alter ‘do not have to be administered in the hospital where heparin is required’ to read ‘can be administered outside the hospital’.

Response 2: Yes, low molecular weight heparins have fewer side effects and require less blood, which means a lower monitoring frequency is needed. This has been corrected in the revised manuscript. (Line 34-35, words highlighted in yellow).According to the reviewer’s suggestions, we have changed “do not have to be administered in the hospital where heparin is required” to “can be administered outside the hospital” in the revised manuscript. (Line 35-36, words highlighted in yellow).

Point 3: In section 2, please start a new numbered paragraph for each preparation method, 2.1 Nitrite degradation; 2.2 Chemical b-elimination; 2.3 Enzymatic b-elimination 2.4 Peroxide degradation, then 2.5 Novel methods for heparin depolymerisaton.

Response 3: According to the reviewer’s suggestions, we have replaced each preparation method with a new numbered paragraph in section 2. (Line 57, 72, 77, 85 and 95, words highlighted in yellow).

Point 4: Lines 77 to 78 Please cite the source of the statement ‘heparanase III does not destroy the core pentasaccharide structure of heparin, so the obtained product has higher anticoagulant activity’.

Response 4: The paper (Org Lett 2018, 20, 48-51) has been suitably cited in the revised manuscript(Line 81, words highlighted in yellow). They are ref. [11].

Point 5: Lines 177, 185 and elsewhere: the term ‘heparin sulfate’ has no clear meaning, please change to either ‘heparan sulfate’ or ‘heparin’

Response 5: According to the reviewer’s suggestions, we have carefully examined the whole manuscript, and “heparin sulfate” has been altered to “heparin”(Lines 196 and 242) and “heparan sulfate”.(Line 187, words highlighted in yellow).

Point 6: Line 258: ‘P-selection’ should be ‘P-selectin’; Line 260: ‘E-selection’ should be ‘E-selectin’ Line 261, line 262: ‘L-selection’ should be ‘L-selectin’.

Response 6: We thank the reviewer for pointing out this important issue. We have corrected the above issues in the revised manuscript. (Line 270, 273, 275, 276 and 277, words highlighted in yellow).

Point 7: Line 386: Please change ‘used instead’ to ‘used interchangeably’.

Response 7: According to the reviewer’s suggestions, we have changed ‘used instead’ to ‘used interchangeably’. (Line 467, words highlighted in yellow).

Point 8: Lines 390-391: Please add the word ‘potentially’ to read ‘enzymatic chemical synthesis has potentially solved the problem of shortage of heparin materials mainly from pig intestinal mucosa’. At present chemoenzymatically synthesized heparins are not available in sufficient quantities or at low enough cost, and their efficacy and safety have not been demonstrated.

Response 8: We agree with the reviewer that our expression is not rigorous enough, and we have added ‘potentially’ to the sentence. (Line 472, words highlighted in yellow).

Point 9: Lines 406-407 Please alter ‘Gamze et al developed a nanometer preparation of enoxaparin’ to read  ‘Ozemre et al. explored nanodelivery systems for enoxaparin’

Response 9: According to the reviewer’s suggestions, relevant modifications have been made in the revised manuscript. (Line 488-489, words highlighted in yellow).

Point 10: In the final section on new drug delivery systems it might be good to mention inhalation of nebulized heparin to treat lung conditions including SARS-CoV-2: for example see Eder et al (mBio. 2022 Dec 20;13(6):e0255822. doi: 10.1128/mbio.02558-22) and the recent review by Shute  (Pharmaceuticals 2023, 16(4), 584; https://doi.org/10.3390/ph16040584).

Response 10: We agree with the reviewer that we have ignored citing some critical work. The paper (Mbio 2022, 13, doi: 10.1128/mbio.02558-22) and (Pharmaceuticals 2023, 16, 584; doi.org/10.3390/ph16040584) have been suitably cited in the revised manuscript. (Line 492 and 495, ref. [128] and [129]).

Reviewer 3 Report

This review on non-anticoagulant properties of LMWHs involves a survey of applications outside of the traditional anticoagulant properties known for these molecules. The survey encompasses a good number of applications including anti-inflammatory, anti-infective, anti-angiogenesis, anti-cell adhesion, anti-fibrotic, etc. This is a good compilation for majority of applications, except for cancer. LMWHs and heparins in general do exhibit very interesting properties can be theoretically lead to their approval for those indications but none have been approved so far. In fact, although the authors appear to indicate that heparins solved some of the issues with COVID (anti-infective), it is important to note that heparins have not been approved for that indication. Also, the authors seem to gloss over much work in the area of anti-cancer activity of heparins and heparin-like molecules (oligosaccharides). Excellent studies showing the value of heparins and heparin-like molecules have been published, which if added to this review, the value of such an article will increase significantly. There are also some wrong statements, several errors and grammatical typos that the authors should revise. Here are the specific suggestions.

1)      Authors should add an entire section on heparins and heparin-like molecules (heparin oligosaccharides, low molecular weight heparins, etc.) in the potential treatment cancer. There are a number of specific clinical trials in progress including ODSH in pediatric cancer patients (NCT02164097); Tinzaparin in cancer (NCT00475098); PG545 in solid tumors (NCT02042781). There are also papers on the subject but one paper that stands out well is heparin in treatment of AML (PMID: 29467192). There are also papers on the subject of heparin hexasaccharide inhibition of cancer stem cells (PMID: 27705927) and its very novel mechanism of inhibition (PMID: 36205924). There are probably other papers and NCTs that the authors can find from a good search of literature and clinicaltrials.gov.

2)      The paper is poor in figures. Figure 1 is not really helpful. Figure 2 should be expanded to include additional molecules from comment 1. Figure 3 is important. Additional figures that help a reader understand the subject better would be the molecular mechanism of how heparins function in each of the phenotypes (anti-inflammatory, anti-infective, anti-angiogenesis, anti-cell adhesion, anti-fibrotic, etc). These should be included.

3)      Wrong information in the paper: a) Heparins have not been approved for COVID as stated on line 50. Revise it to state that the use of heparins in COVID have been investigated but no heparin has been formally approved for use. Actually, some trials with off-label use have indicated positive results, which could be part of the review, but this needs to be stated clearly. b) Heparanase is mentioned several times in the preparation of LMWH section. This is wrong. Heparanase have never been used for preparation. It should be heparinase. The two are very different enzymes. Heparanase is from humans; heparinase is from bacteria. Please check thoroughly and revise. c) When mentioning heparin, do not state heparin sulfate (as on line 177). The correct name is heparin (no sulfate). If you have to mention heparan sulfate, then you cannot mention heparan alone (correct name is heparan sulfate).

4)      Mistakes: a) There is a mention of ‘sulfite’ on line 210. Heparin does not have sulfite groups. It should be sulfate. Check other text to ensure that sulfite is not used elsewhere. b) aldehyde alcohol on line 68 is incorrect. Aldehyde is converted into alcohol to remove reactivity.

5)      Typos: a) All generic names should be lower case. Only trademark names can be capitalized. So enoxaparin should be enoxaparin. However, if you have to mention lovenox, which is a trade name, it should start with capital L (Lovenox). Calcium, Sodium do not need capital letters. Likewise other LMWHs being mentioned in the paper.

6)      Grammatical mistakes: a) Line 47. Sentences do not begin with ‘And’. Revise it. b) Line 184/185/186. Glycosaminoglycan and Bradykinin … no capitalization. c) The authors generally use capitalization without much thought. They should check the full text and revise well.

Changes needed as described in the review

Author Response

We would like to thank you for your careful reading, helpful comments, and constructive suggestions, which have significantly improved the presentation of our manuscript. We have carefully considered all comments and revised the manuscript accordingly. In the following section, we summarize our responses to each comment. We hope our revised manuscript can be accepted for publication.

Comments from Reviewer 3:

This review on non-anticoagulant properties of LMWHs involves a survey of applications outside of the traditional anticoagulant properties known for these molecules. The survey encompasses a good number of applications including anti-inflammatory, anti-infective, anti-angiogenesis, anti-cell adhesion, anti-fibrotic, etc. This is a good compilation for majority of applications, except for cancer. LMWHs and heparins in general do exhibit very interesting properties can be theoretically lead to their approval for those indications but none have been approved so far. In fact, although the authors appear to indicate that heparins solved some of the issues with COVID (anti-infective), it is important to note that heparins have not been approved for that indication. Also, the authors seem to gloss over much work in the area of anti-cancer activity of heparins and heparin-like molecules (oligosaccharides). Excellent studies showing the value of heparins and heparin-like molecules have been published, which if added to this review, the value of such an article will increase significantly. There are also some wrong statements, several errors and grammatical typos that the authors should revise. Here are the specific suggestions.

Point 1: Authors should add an entire section on heparins and heparin-like molecules (heparin oligosaccharides, low molecular weight heparins, etc.) in the potential treatment cancer. There are a number of specific clinical trials in progress including ODSH in pediatric cancer patients (NCT02164097); Tinzaparin in cancer (NCT00475098); PG545 in solid tumors (NCT02042781). There are also papers on the subject but one paper that stands out well is heparin in treatment of AML (PMID: 29467192). There are also papers on the subject of heparin hexasaccharide inhibition of cancer stem cells (PMID: 27705927) and its very novel mechanism of inhibition (PMID: 36205924). There are probably other papers and NCTs that the authors can find from a good search of literature and clinicaltrials.gov..

Response 1: According to the reviewer’s suggestions, we added a section on heparins and heparin-like molecules.(Page 12, words highlighted in yellow). Three clinical trials and three papers have been suitably mentioned in the revised manuscript. (Line 454, ref [108] [109] [111], [113] and [125], words highlighted in yellow).

Point 2: The paper is poor in figures. Figure 1 is not really helpful. Figure 2 should be expanded to include additional molecules from comment 1. Figure 3 is important. Additional figures that help a reader understand the subject better would be the molecular mechanism of how heparins function in each of the phenotypes (anti-inflammatory, anti-infective, anti-angiogenesis, anti-cell adhesion, anti-fibrotic, etc). These should be included.

Response 2: According to the reviewer’s kind suggestion, we have deleted Figure 1. Then we added the structure of PG545 from comment 1 to help readers understand the subject better(Figure 5, line 445). In addition, we have added a picture of the relationship between LMWHs and tumors, including anti angiogenesis and anti-cell adhesion(Figure 3, line 247).

Point 3: Wrong information in the paper: a) Heparins have not been approved for COVID as stated on line 50. Revise it to state that the use of heparins in COVID have been investigated but no heparin has been formally approved for use. Actually, some trials with off-label use have indicated positive results, which could be part of the review, but this needs to be stated clearly. b) Heparanase is mentioned several times in the preparation of LMWH section. This is wrong. Heparanase have never been used for preparation. It should be heparinase. The two are very different enzymes. Heparanase is from humans; heparinase is from bacteria. Please check thoroughly and revise. c) When mentioning heparin, do not state heparin sulfate (as on line 177). The correct name is heparin (no sulfate). If you have to mention heparan sulfate, then you cannot mention heparan alone (correct name is heparan sulfate).

Response 3: According to the reviewer’s kind suggestion, we have carefully examined the whole manuscript and the misrepresentation has been corrected in the revised manuscript.

  1. We have made the expression more rigorous and added the words ”at the experimental level”. Then we added “LMWHs have been investigated in COVID-19, but the use of them have not been officially approved”.(Line 50-53, words highlighted in yellow)
  2. b) Based on the professional suggestions of the reviewers, we have corrected the errors in the entire text and ensured the accurate expression of the term heparinase. (Lines 79, 108 and last line in Table 1 , words highlighted in yellow)
  3. c) We have carefully examined the whole manuscript and “heparin sulfate” has been altered to “heparin”(Lines 196 and 242), and “heparan sulfate”.(Line 187, words highlighted in yellow).

Point 4: Mistakes: a) There is a mention of ‘sulfite’ on line 210. Heparin does not have sulfite groups. It should be sulfate. Check other text to ensure that sulfite is not used elsewhere. b) aldehyde alcohol on line 68 is incorrect. Aldehyde is converted into alcohol to remove reactivity.

Response 4: According to the reviewer’s kind suggestion, we have carefully examined the whole manuscript and the mistakes have been corrected in the revised manuscript.

  1. a) We have changed “sulfite” to “sulfate”.(Line 222)
  2. b) We have changed “aldehyde alcohol” to “alcohol”.(Line 68)

Point 5: Typos: a) All generic names should be lower case. Only trademark names can be capitalized. So enoxaparin should be enoxaparin. However, if you have to mention lovenox, which is a trade name, it should start with capital L (Lovenox). Calcium, Sodium do not need capital letters. Likewise other LMWHs being mentioned in the paper.

Response 5: According to the reviewer’s suggestions, we have carefully examined the whole manuscript and the types have been corrected in the revised manuscript (Line 46, 47, 70, 75, 76, 83, 91, 93, 94, 103, 109, 110, 165, 176, 195, 197, 243, 244, 364, 449, 452, Table 1 and 2, words highlighted in yellow).

Point 6: Grammatical mistakes: a) Line 47. Sentences do not begin with ‘And’. Revise it. b) Line 184/185/186. Glycosaminoglycan and Bradykinin … no capitalization. c) The authors generally use capitalization without much thought. They should check the full text and revise well.

Response 6: Thanks for the professional suggestions of the reviewers.

  1. We have deleted “And”. (Line 47, words highlighted in yellow).
  2. “Glycosaminoglycan and Bradykinin” have been changed to “glycosaminoglycan and bradykinin”. (Line 195, words highlighted in yellow).
  3. According to the reviewer’s suggestions, we checked the full text and revised it.

(Line 46, 47, 70, 75, 76, 83, 91, 93, 94, 103, 109, 110, 165, 176, 195, 197, 243, 244, 364, 449, 452, Table 1 and 2, words highlighted in yellow).

Round 2

Reviewer 1 Report

Accept in present form

Minor English formatting errors should be corrected.